# Exploring cardiovascular disease prevalence and contributing factors among adults in Southern Iran, a cross-sectional survey: Rationale, design, and primary results

Hossein Farshidi[1], Marzieh Nikparvar[1], Shokrollah Mohseni[2], Ebrahim Eftekhar[3], Farideh Dastsouz[1], Abdoljabbar Zakeri[1], Mohammad Mohammadi[4], Farkhondeh Razmpour[4], Farid Khorrami[5], Masoumeh Mahmoodi[1], Nahid Shahabi[1], Soghra Fallahi[1], Atefeh Ghareghani[1], Najmeh Shanbehzadeh[6], Shahin Abbaszadeh [1]*

**1** Cardiovascular Research Center, Hormozgan University of Medical Sciences, Bandar Abbas, Iran, **2** Social Determinants in Health Promotion Research Center, Hormozgan Health Institute, Hormozgan University of Medical Sciences, Bandar Abbas, Iran, **3** Molecular Medicine Research Center, Hormozgan Health Institute, Hormozgan University of Medical Sciences, Bandar Abbas, Iran, **4** Food Health Research Center, Hormozgan University of Medical Sciences, Bandar Abbas, Iran, **5** Health Information Technology, Faculty of Paramedicine, Hormozgan University of Medical Sciences, Bandar Abbas, Iran, **6** Department of Oral and Maxillofacial Medicine, School of Dentistry, Hormozgan University of Medical Sciences, Bandar Abbas, Iran

* sh.abbaszadeh@hums.ac.ir

## Abstract

### Background and objectives

Cardiovascular disease (CVD) is the primary cause of death worldwide, with significant fatalities reported in Iran. Hormozgan Province in southern Iran faces serious public health challenges related to CVD, a major cause of morbidity and mortality in the region. This study investigates the prevalence of CVD and its contributing factors in this region.

### Method

This cross-sectional study was conducted in Hormozgan Province, southern Iran, from June 21 to December 21, 2023. A total of 6289 individuals aged 20 and older were selected through random multistage sampling techniques. Data collection involved three phases: administering a comprehensive questionnaire, measuring anthropometric data and blood pressure, and conducting laboratory blood tests. CVD was defined as the presence of ischemic heart disease (IHD) or ischemic and hemorrhagic strokes. Data analysis was performed using SPSS version 26, incorporating statistical tests such as multivariable logistic regression, Mann-Whitney tests, and Chi-square to compare groups.

**Data availability statement:** All relevant data are within the paper and its Supporting Information files.

**Funding:** This study was supported by Hormozgan University of Medical Sciences, Iran. The funding organization had no influence on the study's design, data collection, analysis, interpretation of the data, or the writing of the manuscript.

**Competing interests:** The authors declare no competing interests.

**Abbreviation:** CVD: Cardiovascular Disease; IHD: Ischemic Heart Disease; BMI: Body Mass Index; HTN: Hypertension; DALYs: Disability-Adjusted Life Years; WHO: World Health Organization; STEPS: Stepwise Approach to Non-Communicable Disease Risk Factor Surveillance; GPAQ: The Global Physical Activity Questionnaire; HADS: Hospital Anxiety and Depression Scale; WHR: Waist-To-Hip Ratio; SBP: Systolic Blood Pressure; DBP: Diastolic Blood Pressure; CBC: Complete Blood Count; FBS: Fasting Blood Sugar; TC: Total Cholesterol; HDL-C: High-Density Lipoproteins Cholesterol; LDL-C: Low-Density Lipoproteins Cholesterol; TG: Triglycerides; ALT: Alanine Aminotransferase; AST: Aspartate Aminotransferase; BUN: Blood Urea Nitrogen; Cr: Creatinine; TSH: Thyroid-Stimulating Hormone; CRP: C-Reactive Protein; Pre-HTN: Pre Hypertension; COR: Crude Odds Ratios; AOR: Adjusted Odds Ratios; Cis: Confidence Intervals; WBC: White Blood Cell Count; MENA: Middle East and North Africa.

## Results

A total of 6289 participants were included in the study, with a mean age of 45.63 ± 15.04 years. Among them, 58.5% were female, and 51.6% lived in urban areas. Key findings revealed that 45.7% of the population were identified as overweight or obese, with a significant prevalence of abdominal obesity at 69.6%, particularly among females. The prevalence of CVD in the population was 7.8%, with IHD affecting 6.7% of individuals and stroke affecting 2.2%. Multivariable logistic regression identified increased age (AOR: 1.03; 95% CI: 1.02–1.04), higher body mass index (BMI) (AOR: 1.02; 95% CI: 1.00–1.05), hypercholesterolemia (AOR: 1.30; 95% CI: 1.03–1.66), hypertension (HTN) (AOR: 2.02; 95% CI: 1.55–2.64), diabetes (AOR: 1.31; 95% CI: 1.00–1.71), and severe anxiety (AOR: 2.39; 95% CI: 1.30–4.39) as significant risk factors for CVD. Women had a 33% lower risk of having CVD compared to men (AOR: 0.67; 95% CI: 0.53–0.85).

## Conclusion

This study highlights urgent public health concerns in Hormozgan Province, including high rates of CVD, obesity, abnormal blood pressure, and diabetes, particularly among females, underlining the need for targeted health interventions and improved nutritional practices.

## Introduction

Cardiovascular disease (CVD) is the leading cause of death worldwide, encompassing a range of conditions such as coronary heart disease, cerebrovascular disease, and rheumatic heart disease [1,2]. In 2021 alone, CVD was responsible for approximately 16.2 million fatalities [1], with heart attacks and strokes accounting for over 80% of these deaths [3]. Estimates predict that this number could escalate to 35.6 million by 2050 [4]. Moreover, in 2022, CVD contributed to more than 344 million disability-adjusted life years (DALYs), making it one of the most burdensome health challenges globally, with this figure expected to surpass 365 million by 2050 [5].

In Iran, CVD has been a significant public health problem and having been the leading cause of mortality from 2010 to 2019 [1,6]. Although the COVID-19 pandemic temporarily shifted the mortality landscape—making it the primary cause of death in 2021—CVD still accounted for 16.1 deaths per 100,000 people, underscoring its enduring impact on public health [1]. In addition, from 1990 to 2019, CVD consistently ranked as the leading cause of age-standardized DALY rates in this area [6].

The rising prevalence of CVD can be attributed to demographic shifts, including population growth and aging, particularly in regions such as Western Asia, Northern Africa, and parts of Latin America and the Caribbean, where the elderly population is projected to double by 2050 [7–9]. Iran's reported prevalence of 8.11% for CVD in 2021, therefore identifies a priority health issue and is consistent with overall global patterns [10].

While the relationship between specific risk factors and CVD is well established, it remains crucial to identify high-risk individuals and analyze trends in prevalence alongside demographic, medical, socioeconomic, and lifestyle factors [11,12]. Key behavioral risk factors—such as poor dietary habits, smoking, physical inactivity, and excessive alcohol consumption—often lead to conditions like hypertension (HTN), dyslipidemia, hyperglycemia, and obesity [3,13–15]. Socioeconomic status significantly influences CVD incidence and mortality, with lower-income groups experiencing disproportionately higher rates of CVD, emphasizing the need to understand the socioeconomic determinants of health [16].

Hormozgan Province, located in southern Iran with a population of around 2 million, exemplifies the evident need for targeted interventions. In 2011, ischemic heart disease (IHD) was the leading cause of mortality in this region; however, by 2021, COVID-19 caused more deaths than IHD. Despite this shift, stroke and IHD remain critical contributors to mortality, with high blood pressure emerging as the primary driver of DALYs in Hormozgan [17].

Understanding these evolving dynamics is crucial for devising effective and targeted interventions. By updating our knowledge of the CVD prevalence and the trends in risk factors, this study aims to inform public health professionals and researchers worldwide about the current state of CVD in southern Iran, and inspire effective recommendations and further research for comparative analysis- ultimately enhancing public health strategies in this economically vital region of Iran and contributing to the global discourse on CVD and its determinants.

## Materials and methods

### Design and geographic location

This cross-sectional study was conducted in Hormozgan Province, Southern Iran, between June 21 and December 21, 2023.

Participants were recruited from 13 counties in Hormozgan: Bandar Abbas, Minab, Rudan, Sirik, Jask, Bashagard, Bandar Khamir, Hajjiabad, Qeshm, Bandar Lengeh, Bastak, Abumousa, and Parsian. Each county has its unique geography, culture, and economy, contributing to the province's rich diversity.

### Ethics approval and consent to participate

This study received approval from the Ethics Committee of Hormozgan University of Medical Sciences (IR.HUMS. REC.1401.080). Informed consent was obtained from all participants prior to their involvement in the study.

**Study participants.** Eligible participants included males and females aged 20 years and older who had resided in Hormozgan Province for at least one year prior to the start of the study. Participants were asked to provide written informed consent before taking part in the study. Pregnant women, individuals with specific medical conditions (such as active cancer treatment, liver or renal failure, major surgery within the past 3 months), and those with physical or mental disabilities that hindered their ability to enroll were excluded from the study (**Fig 1**).

**Sample size and selection.** The sample size was calculated using the proportion estimation formula. To ensure that the sample size would be adequate and representative for estimating various risk factors, we selected a value of P = 0.50. Cluster sampling techniques were employed, and a design effect of 1.5 was used to account for the sampling structure. Additionally, we considered five age groups, included a gender variable, and anticipated a non-response rate of 10–12%. Accordingly, the final estimated sample size was calculated to be 6504 participants.

$$n_1 = \frac{z^2_{1-\frac{a}{2}} \times pq}{d^2} = \frac{1.96^2 \times (0.5)(0.5)}{0.05^2} = 384.16$$

$$n_2 = 384.16 \times 1.5 \times 10 = 5762.4$$

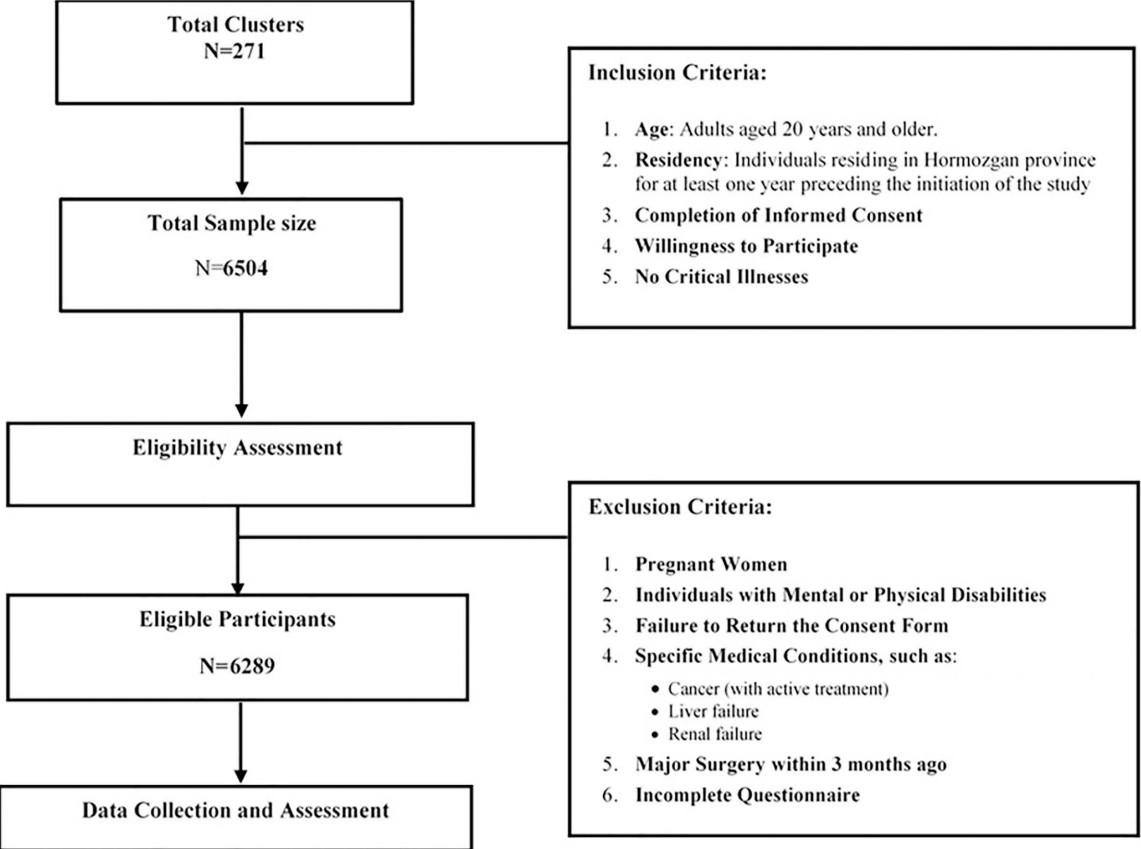

**Fig 1. Sampling and recruitment flowchart.**

$$n_3 = 5762.4 \div (1 - 0.88) \approx 6504$$

A multi-stage sampling method was used to obtain an adequate sample of adults aged 20 years and older in Hormozgan Province. This approach combined stratified, cluster, and random sampling techniques, as described below:

1. **Stratification by county (first stage):** Each county in Hormozgan Province was considered a stratum. The sample size was distributed proportionally according to the population of individuals aged 20 years and older in each county.

2. **Stratification of urban and rural regions (stage II):** Each county was further divided into urban and rural regions. Within each county, the sample allocation was proportional to the adult populations in the urban and rural regions.

3. **Cluster definition and selection (third stage):** Operationally, clusters were defined as units of health services. In the urban regions, clusters were Comprehensive Health Service Centers or Health Bases; in the rural regions, clusters were Health Service Units. The clusters from each stratum (urban or rural within each county) were distributed proportionally to the population of that stratum. Clusters were chosen by simple random sampling from the list of eligible health services units.

4. **Recruitment of clusters and participants in clusters:** Each cluster had 24 participants. All of the selected health units had households enrolled, listed by the national identification codes of the household heads. From these, an initial

("cluster head") household was randomly selected. Data collection began with this household and proceeded sequentially to neighboring households (usually moving to the household immediately on the right) until the cluster quota was filled. All individuals aged 20 years and older in the selected households were visited.

5. **Substitution for non-responding households:** If the selected household was not present at the first visit, a notice was left at the house. Two additional visits were made at 24- and 72-hour intervals. If the household was still not available, the next eligible household to the right was chosen as a replacement.

The sampling methods employed for cluster selection are presented in **Table 1**.

**Data collection and assessment.** The study followed a structured three-step approach: (1) questionnaire completion, (2) anthropometric and blood pressure measurements, and (3) laboratory tests.

**Step 1: Questionnaire completion.** Skilled and trained interviewers utilized tablets to facilitate the completion of questionnaires. Prior to the study, a training workshop was held to instruct interviewers on effective techniques for administering the questionnaire. Each interviewer introduced themselves and explained the purpose of the study, then provided participants with a written consent form for their approval. The questionnaire comprised 13 sections, totaling 181 questions. The questionnaires were then completed using tablets. At the end of each week, the data collected by the interviewers was monitored and evaluated for errors so that necessary corrections could be made.

These sections are organized as follows:

1. Cluster and household information: 12 questions

2. Ecological data: 12 questions (age, gender, place of residence, education, occupation status, and family size)

3. Family wealth: 12 questions

4. Physical activity: 17 questions

5. HTN: 9 questions

6. Diabetes: 22 questions

**Table 1. Sample selection overview: urban and rural areas.**

| Counties | Total Population (>20 y) | Urban population (>20 y) | Rural population (>20 y) | Urban (%) | Clusters (N) | Urban cluster (N) | Rural cluster (N) | Calculated sample size (N) |
|---|---|---|---|---|---|---|---|---|
| Bandar-Abbas | 399556 | 279689 | 119867 | 70 | 100 | 70 | 30 | 2400 |
| Minab | 162232 | 56781 | 105451 | 35 | 40 | 14 | 26 | 960 |
| Rudan | 80248 | 32099 | 48149 | 40 | 20 | 8 | 12 | 480 |
| Sirik | 31024 | 7756 | 23268 | 25 | 8 | 2 | 6 | 192 |
| Jask | 34612 | 10383 | 24229 | 30 | 10 | 3 | 7 | 240 |
| Bashagard | 18064 | 5961 | 12103 | 33 | 6 | 2 | 4 | 144 |
| Hajjiabad | 38876 | 19438 | 19438 | 50 | 10 | 5 | 5 | 240 |
| Qeshm | 87201 | 43600 | 43601 | 50 | 22 | 11 | 11 | 528 |
| Bandar- Khamir | 36189 | 16285 | 19904 | 45 | 9 | 4 | 5 | 216 |
| Bandar-Lengeh | 101623 | 42681 | 58942 | 42 | 24 | 10 | 14 | 576 |
| Bastak | 51131 | 15850 | 35281 | 31 | 13 | 4 | 9 | 312 |
| Parsian | 32903 | 20399 | 12504 | 62 | 8 | 5 | 3 | 192 |
| Abumusa | 1112 | 1112 | 0 | 100 | 1 | 1 | 0 | 24 |
| **Total** | **1074771** | **548133** | **526638** | **51** | **271** | **139** | **132** | **6504** |

7. Cholesterol: 8 questions

8. History of CVD: 8 questions

9. Tobacco use status: 37 questions

10. Alcohol consumption: 12 questions

11. Lifestyle recommendations: 7 questions

12. Anthropometric and blood pressure measurements: 11 questions

13. Depression and anxiety: 14 questions

The framework for the questionnaires was adapted from the World Health Organization's (WHO) stepwise approach to non-communicable disease risk factor surveillance (STEPS) survey. Additional elements were incorporated to align with the objectives of our study. All questionnaires utilized in the present study have been appropriately tested for validity and reliability in the Iranian population [10,18,19].

The second version of the Global Physical Activity Questionnaire (GPAQ), developed by the WHO, was used to evaluate physical activity levels and sedentary behaviors [19,20]. One MET is defined as the energy expenditure of one kilocalorie per kilogram of body weight per hour when a person is at rest [21]. A total of more than 600 MET-minutes per week was considered sufficient physical activity [22].

Education was quantified based on the number of years of formal schooling that individuals had successfully completed, categorized into six distinct groups: less than one year (illiteracy), 1 to 5 years (primary school), 6 to 8 years (middle school), 9 to 12 years (high school), 13 to 16 years (bachelor's degree), and more than 16 years (master's/PhD degree).

To evaluate participants' socioeconomic status, we utilized principal component analysis to develop a wealth index. The variables considered in determining the wealth index included the type of residence, ownership status (owned or rented), construction materials, type of cooking fuel, source of lighting, household items and possessions, water supply for domestic use, and the type of sanitation facilities available. Data for this analysis were derived from the WHO's STEPS survey, which provides standardized tools for collecting and analyzing data on various health-related factors [23]. The resulting wealth indices were categorized into five quintiles, ranging from the lowest (first quintile, indicating the poorest individuals) to the highest (fifth quintile, signifying the wealthiest).

Participant stress and anxiety levels were assessed using the Hospital Anxiety and Depression Scale (HADS), a validated 14-item self-report measure that screens for symptoms of depression and anxiety in both clinical and general populations. The HADS consists of 14 items divided evenly into two subscales: seven items assess anxiety symptoms, and seven items assess depression symptoms. Each item is rated on a 4-point Likert scale (0 to 3), resulting in subscale scores ranging from 0 to 21. Subscale scores are categorized as follows: 0–7, none or slight; 8–10, mild; 11–14 moderate; and 15–21, severe [18,24].

**Step 2: Anthropometric and blood pressure measurements.** After participants completed the questionnaires, anthropometric measurements were obtained following standardized protocols. Weight was measured using a digital scale (Imperial) with participants dressed in light clothing and barefoot, ensuring accuracy to within 100 grams. Height was recorded with a non-stretchable tape measure as participants stood barefoot with their heels, back, and head against a wall, looking straight ahead. Body Mass Index (BMI) was calculated as weight in kilograms divided by the square of height in meters. Waist circumference was measured using a non-stretchable tape measure precise to 0.1 cm, without applying pressure, at the midpoint between the last rib and the top of the hip bone (iliac crest). Hip circumference was measured at the widest point of the hips. subsequently, the Waist-to-Hip Ratio (WHR) was calculated. Systolic and diastolic blood pressure (SBP, DBP) were assessed using a digital monitor following a minimum of 15 minutes of rest. Three readings

were taken at five-minute intervals from the right arm, and the average value was used for analysis to ensure accuracy. Participants were instructed to refrain from smoking, alcohol, coffee, stimulants, intense physical activity, and heavy meals for at least 30 minutes before measurements.

**Step 3: Laboratory tests.** Participants were then referred to the laboratory for further tests and were asked to schedule their visit within 48 to 72 hours after completing the questionnaire. A 12-hour fasting period was required to ensure accurate results. A comprehensive assessment was conducted, including analysis of the complete blood count (CBC), total cholesterol (TC), Low-Density Lipoproteins cholesterol (LDL-C), High-Density Lipoproteins cholesterol (HDL-C), fasting blood sugar (FBS) level, triglycerides (TG), liver enzymes (aspartate transaminase (AST), alanine transaminase (ALT), γ-glutamyl transpeptidase, alkaline phosphatase), kidney function tests (blood urea nitrogen (BUN), creatinine (Cr)), serum Thyroid-Stimulating Hormone (TSH). Fasting blood samples were collected into 10 ml vacuum tubes with a clot activator (Parspayvand Company, Iran) between 8:00 and 9:00 a.m. All samples were allowed to clot for 30 minutes at room temperature before being centrifuged at 3000 rpm for 10 minutes. The serum was separated and aliquoted into 1.5 mL microtubes for biochemical analysis. The first-morning urine was used for microscopic and macroscopic urine analysis as well as for urine albumin and creatinine assay. Serum lipid profile (TG, TC, LDL-C, and HDL-C), FBS, ALT, AST, BUN, Cr, urine albumin, and urine creatinine were measured on the Mindray BS-200 fully automated biochemistry analyzer (Biotechnical Instruments, China) using commercially standard kits (Audit, Delta Darman Part Company, Tehran, Iran). Serum C-reactive protein (CRP) was determined on a BT-1500 chemistry autoanalyzer (Biotechnical Instruments, Rome, Italy) using a standard kit (Bionik, Tehran, Iran). Serum sodium and potassium levels were measured using an EasyLytePlus electrolyte analyzer (Medica Company, USA). Whole blood (K2EDTA 1 mg/ml) was used for HbA1C assay (Rayton Kit, Tehran, Iran) and CBC analysis. Serum TSH levels were determined using the ELISA method (Pishtaz Teb ELISA kit, Tehran, Iran) according to the manufacturer's instructions. The intra-assay and inter-assay CV for the kit were 6% and 8%, respectively.

## Operational definition

CVD in this study was defined as the presence of IHD or ischemic or hemorrhagic stroke. The CVD events were initially self-reported by the participants through questionnaires. The medical records were then processed and confirmed by a cardiologist for the diagnosis. For stroke cases, the diagnosis was confirmed based on clinical examination and imaging findings documented in the medical records. Confirmed events were then counted as participants outcomes. The WHR is a key indicator of abdominal obesity and is associated with an increased risk of conditions such as CVD and diabetes. According to WHO guidelines, abdominal obesity is defined by a WHR exceeding 0.90 for males and 0.85 for females [25]. Participants were categorized based on their BMI into the following classifications:

- Underweight: <18.5 kg/m²

- Normal weight: 18.5 to 24.9 kg/m²

- Overweight: 25 to 29.9 kg/m²

- Obese: ≥ 30 kg/m²

Diabetes is diagnosed when FBS levels are 126 mg/dL or higher, or if the individual is currently on medication. Prediabetes is defined by FBS levels between 100 and 125.9 mg/dL in the absence of medication [26]. The criteria for successful diabetes management include maintaining an FBS level below 130 mg/dL and a hemoglobin A1c level of less than 7% [27]. HTN is identified when measurements indicate SBP is ≥ 140 mmHg or DBP is ≥ 90 mmHg, or if the person is currently taking antihypertensive medication. Prehypertension (Pre-HTN) is characterized by SBP between 120 and 139.9 mmHg or DBP between 80 and 89.9 mmHg without medication [28]. The criteria for successful HTN therapy were defined as achieving a blood pressure below 140/90 mmHg [29]. Hypercholesterolemia is defined by TC levels of 200 mg/dL or more, or if the individual is currently on medication [30].

## Data analysis tools and methods

All statistical analyses were conducted using SPSS version 26. Quantitative variables were described using means and standard deviations, while categorical and qualitative variables were represented by frequency counts and percentages. The Chi-square test was employed to compare the general characteristics of subjects between the two groups for qualitative variables. The Kolmogorov-Smirnov test was used to evaluate the normality of the data distribution. The Mann-Whitney test was applied to quantitative variables that did not exhibit a normal distribution. Univariate and multivariable logistic regressions were conducted to identify associations between various characteristics and metabolic variables with CVD. Variables that yielded a p-value less than 0.2 in the univariate analysis were included in the multivariable logistic regression model [31]. In the multivariable model, we systematically removed variables with the highest p-values until all remaining variables had p-values below 0.05. Both crude odds ratios (COR) and adjusted odds ratios (AOR), along with their 95% confidence intervals (CIs), were estimated to determine the strength of association between independent variables and CVD. A p-value less than 0.05 was considered statistically significant.

## Results

The sample size was 6,504, of whom 6,289 completed the questionnaire, resulting in a high response rate of 96.6%. The mean age of the participants was 45.63 ± 15.04 years. Among the responders, 2,613 (41.5%) were male, and 3,676 (58.5%) identified as female. Furthermore, 3,248 individuals (51.6%) lived in urban settings, whereas 3,041 (48.4%) resided in rural areas. In terms of marital status, the population consisted of 4,960 (78.9%) married individuals. Only 672 (10.7%) reported low literacy levels or minimal educational attainment. Additionally, 464 (7.4%) of the population were unemployed (**Table 2**). Statistically significant variation was observed for age, education level, marital status, occupation, and wealth index when comparing the individuals by gender or residence location.

The anthropometric data and laboratory results of the participants in the study are given in **Table 3**. The average BMI was found to be 24.9 ± 4.7. SBP and DBP readings averaged 118 ± 15.6 mmHg and 74 ± 10.3 mmHg, respectively. The average FBS was considerably high at approximately 109 ± 43.6 mg/dL. It is important to highlight that all variables, except for waist circumference, white blood cell (WBC), TC, and LDL-C levels, exhibited statistically significant differences among male and female subjects (p < 0.05).

Table 4 shows the prevalence of CVD and various risk factors, such as obesity, physical inactivity, smoking, diabetes, HTN, hypercholesterolemia, depression, and anxiety. The results are classified according to gender and residence. The overall prevalence of CVD in Hormozgan Province is 489 (7.8%), and there were no gender or urban/rural differences. Among those affected, 422 (6.7%) have experienced IHD, while 136 (2.2%) have suffered from a stroke. A significant difference in the prevalence of IHD was observed between males (197; 7.5%) and females (225; 6.1%) (P = 0.027). In addition, 73 (14.9%) of the individuals with a history of CVD reported a family history of the condition among their first-degree relatives.

Among the studied population, 45.7% were classified as overweight or obese, with a higher prevalence of both conditions observed in females and urban areas. The prevalence of abdominal obesity, as indicated by the WHR, was significantly high at 69.6% of the population, and a greater prevalence was noted in females than in males. Additionally, 57.6% of individuals reported insufficient physical activity, with a higher prevalence of inactivity among females. Regarding smoking habits, the overall prevalence of smoking within the community was 10.3%; hookah smoking was more common among women, while cigarette smoking was more prevalent among men. The prevalence of HTN and pre-HTN in the population was 24% and 39.1%, respectively, indicating that 63.1% of the population have abnormal blood pressure levels. Notably, the prevalence of HTN was higher in females, whereas pre-HTN was more prevalent in males. In Hormozgan Province, the rates of diabetes and prediabetes

**Table 2. Demographic and socioeconomic characteristics of participants.**

| Variable | Total (N = 6289) | Gender | | P value | Residence | | P value[2] |
|---|---|---|---|---|---|---|---|
| | | Male (N = 2613) | Female (N = 3676) | | Urban (N = 3248) | Rural (N = 3041) | |
| **Age (years)** | | | | | | | |
| 20-29 | 902 (14.3) | 363 (13.9) | 539 (14.7) | | 480 (14.8) | 422 (13.9) | |
| 30-39 | 1596 (25.4) | 665 (25.4) | 931 (25.3) | | 854 (26.3) | 742 (24.4) | |
| 40-49 | 1444 (23) | 584 (22.3) | 860 (23.4) | 0.045 | 799 (24.6) | 645 (21.2) | <0.001 |
| 50-59 | 1069 (17) | 424 (16.2) | 645 (17.5) | | 516 (15.9) | 553 (18.2) | |
| ≥ 60 | 1278 (20.3) | 577 (22.1) | 701 (19.1) | | 599 (18.4) | 679 (22.3) | |
| **Education** | | | | | | | |
| Illiteracy (<1) year | 672 (10.7) | 232 (8.9) | 440 (12) | | 236 (7.3) | 436 (14.3) | |
| Primary school (1–5) | 828 (13.2) | 363 (13.9) | 465 (12.6) | | 391 (12) | 437 (14.4) | |
| Secondary school (6–8) | 1744 (27.7) | 674(25.8) | 1070 (29.1) | <0.001 | 785 (24.2) | 959 (31.5) | <0.001 |
| High school (9–12) | 2029 (32.3) | 844 (32.3) | 1185 (32.2) | | 1072 (33) | 957 (31.5) | |
| Bachelor's degree (13–16) | 933 (14.8) | 456 (17.5) | 477 (13) | | 694 (21.4) | 239 (7.9) | |
| Masters/PhD degree (>16) | 83 (1.3) | 44 (1.7) | 39 (1.1) | | 70 (2.2) | 13 (0.4) | |
| **Marital status** | | | | | | | |
| Single | 768 (12.2) | 346 (13.2) | 422 (11.5) | | 414 (12.7) | 354 (11.6) | |
| Married | 4960 (78.9) | 2204 (84.3) | 2756 (75) | <0.001 | 2594 (79.9) | 2366 (77.8) | <0.001 |
| Divorce/Widow | 561 (8.9) | 63 (2.4) | 498 (13.5) | | 240 (7.4) | 321 (10.6) | |
| **Occupation** | | | | | | | |
| Employed | 2281 (36.3) | 1843 (70.5) | 438 (11.9) | | 1274 (39.2) | 1007 (33.1) | |
| Unemployed | 464 (7.4) | 340 (13) | 124 (3.4) | <0.001 | 227 (7) | 237 (7.8) | <0.001 |
| Housewife | 3248 (51.6) | 184 (7) | 3064 (83.4) | | 1538 (47.4) | 1710 (56.2) | |
| Retired | 296 (4.7) | 246 (9.4) | 50 (1.4) | | 209 (6.4) | 87 (2.9) | |
| **Wealth index** | | | | | | | |
| Quintile 1 (Poorest) | 1261 (20.1) | 431 (16.5) | 830 (22.6) | | 469 (14.4) | 792 (26) | |
| Quintile 2 | 1259 (20) | 467 (17.9) | 792 (21.5) | | 567 (17.5) | 692 (22.8) | |
| Quintile 3 | 1254 (19.9) | 530 (20.3) | 724 (19.7) | <0.001 | 666 (20.5) | 588 (19.3) | <0.001 |
| Quintile 4 | 1258 (20) | 585 (22.4) | 673 (18.3) | | 709 (21.8) | 549 (18.1) | |
| Quintile 5 (Richest) | 1257 (20) | 600 (23) | 657 (17.9) | | 837 (25.8) | 420 (13.8) | |

[1]Values are reported as frequency (%).

[2]P values computed by Chi-square.

were found to be 20.6% and 32.1%, respectively. The results showed no significant gender variation in the prevalence of hypercholesterolemia, depression, and diabetes. However, prediabetes was more prevalent among men, whereas anxiety was more frequent in women. Higher rates of depression and anxiety were also observed in urban areas.

Fig 2 illustrates the distribution of individuals regarding the prevalence, diagnosis, and control of diabetes and HTN. Treatment success rates were indicated to be 46.6% for diabetes and 64.3% for HTN. The study found that 30.2% of HTN and 36.9% of diabetic subjects were unaware of their conditions. Throughout the study, these subjects became aware of and recognized their diseases.

Univariate and multivariable logistic regression analyses were performed to examine the associations between various characteristics, metabolic variables, and CVD (see Tables 5 and Table 6). The odds ratio of CVD is twofold higher in individuals with HTN (AOR: 2.02; 95%CI: 1.55–2.64) and severe anxiety (AOR: 2.39; 95%CI: 1.30–4.39) compared to those without these conditions. Additionally, individuals with diabetes and hypercholesterolemia had odds ratios of 1.31 (AOR: 1.31; 95%CI: 1.00–1.71) and 1.30 (AOR: 1.30; 95%CI: 1.03–1.66) for CVD, respectively. The odds ratio for CVD in females was 33% lower than in males (AOR: 0.67; 95% CI: 0.53–0.85). Furthermore, each one-unit increase in age and BMI resulted in an increase in odds of CVD, with odds ratios of 1.03 (AOR: 1.03; 95%CI: 1.02–1.04) for age and 1.02 (AOR: 1.02; 95%CI: 1.00–1.05) for BMI.

**Table 3. Anthropometric variables and laboratory data of participants.**

| Variable | Total | Gender | | P value | Residence | | P value[2] |
|---|---|---|---|---|---|---|---|
| | | Male | Female | | Urban | Rural | |
| **Age (year)** | 45.63±15.04 | 46.28±15.69 | 45.15±14.55 | 0.054 | 44.77±14.54 | 46.54±15.50 | <0.001 |
| **Weight (kg)** | 66.68±14.06 | 70.50±14.29 | 63.97±13.24 | <0.001 | 68.45±14.39 | 64.80±13.45 | <0.001 |
| **Height (cm)** | 163.44±8.81 | 169.59±7.87 | 159.07±6.55 | <0.001 | 164.16±8.94 | 162.67±8.61 | <0.001 |
| **BMI (kg/m²)** | 24.92±4.73 | 24.45±4.36 | 25.26±4.94 | <0.001 | 25.35±4.74 | 24.46±4.66 | <0.001 |
| **Waist circumference (cm)** | 88.87±12.02 | 89.15±11.70 | 88.68±12.24 | <0.318 | 89.86±11.85 | 87.82±12.11 | <0.001 |
| **Hip circumference (cm)** | 97.59±12.55 | 96.88±12.31 | 98.10±12.69 | <0.001 | 98.70±12.49 | 96.41±12.51 | <0.001 |
| **WHR** | 0.91±0.10 | 0.92±0.09 | 0.90±0.10 | <0.001 | 0.91±0.08 | 0.91±0.11 | <0.002 |
| **SBP (mmHg)** | 118.36±15.66 | 120.01±14.79 | 117.19±16.14 | <0.001 | 119.36±15.65 | 117.29±15.59 | <0.001 |
| **DBP (mmHg)** | 74.71±10.33 | 75.25±10.07 | 74.33±10.49 | <0.001 | 75.87±10.21 | 73.47±10.30 | <0.001 |
| **Hemoglobin (gr/dl)** | 12.71±1.74 | 13.77±1.69 | 12.11±1.46 | <0.001 | 12.81±1.72 | 12.61±1.75 | <0.002 |
| **Hematocrit (gr/dl)** | 39.88±4.34 | 42.63±4.15 | 38.32±3.62 | <0.001 | 39.96±4.23 | 39.80±4.45 | <0.395 |
| **WBC (thousand/mm³)** | 6.31±1.80 | 6.31±1.85 | 6.31±1.77 | <0.487 | 6.22±1.76 | 6.40±1.83 | <0.002 |
| **RBC (million/mm³)** | 4.97±0.64 | 5.26±0.66 | 4.81±0.57 | <0.001 | 4.94±0.65 | 5.00±0.63 | <0.001 |
| **FBS (mg/dl)** | 109.31±43.64 | 111.12±45.57 | 108.29±42.49 | <0.001 | 109.00±40.99 | 109.59±45.94 | <0.414 |
| **TC (mg/dl)** | 185.72±43.29 | 185.04±44.69 | 186.10±42.48 | <0.410 | 185.18±42.88 | 186.21±43.66 | <0.428 |
| **TG (mg/dl)** | 139.49±96.32 | 155.27±115.64 | 130.59±82.18 | <0.001 | 142.44±103.38 | 136.80±89.32 | <0.304 |
| **LDL-C (mg/dl)** | 112.07±34.85 | 112.42±35.67 | 111.87±34.39 | <0.609 | 110.89±34.95 | 113.14±34.73 | <0.100 |
| **HDL-C (mg/dl)** | 47.59±11.22 | 45.69±10.76 | 48.66±11.33 | <0.001 | 47.62±10.72 | 47.57±11.65 | <0.339 |
| **AST (IU/L)** | 22.32±11.51 | 24.38±11.17 | 21.16±11.54 | <0.001 | 22.38±12.85 | 22.26±10.14 | <0.219 |
| **ALT (IU/L)** | 21.92±17.61 | 26.34±20.29 | 19.43±15.35 | <0.001 | 22.51±18.51 | 21.38±16.72 | <0.012 |
| **TSH (µIU/ml)** | 2.26±2.99 | 2.01±2.47 | 2.40±3.24 | <0.001 | 2.34±3.19 | 2.19±2.80 | 0.001 |
| **BUN (mg/dl)** | 29.22±10.47 | 32.45±10.54 | 27.41±9.99 | <0.001 | 28.49±10.38 | 29.89±10.51 | <0.001 |
| **Cr (mg/dl)** | 0.95±0.24 | 1.07±0.19 | 0.88±0.24 | <0.001 | 0.94±0.24 | 0.95±0.24 | <0.001 |
| **CRP (mg/L)** | 3.66±4.14 | 3.48±4.57 | 3.76±3.88 | <0.001 | 3.49±4.42 | 3.81±3.87 | <0.001 |

[1]Values are reported as Mean±Standard Deviation (SD).

[2]P values computed by Mann–Whitney U-test.

**BMI**: Body Mass Index, **WHR**: Waist-to-Hip Ratio, **SBP**: Systolic Blood Pressure, **DBP**: Diastolic Blood Pressure, **WBC**: White Blood Cells, **RBC**: Red Blood Cells, **FBS**: Fasting Blood Sugar, **TC**: Total Cholesterol, **TG**: Triglycerides, **LDL-C**: Low-Density Lipoproteins, **HDL-C**: High-Density Lipoproteins, **AST**: Aspartate Aminotransferase, **ALT**: Alanine Aminotransferase, **TSH**: Thyroid-Stimulating Hormone, **BUN**: Blood Urea Nitrogen, **Cr**: Creatinine, **CRP**: C- reactive protein.

## Discussion

The increasing prevalence of CVD is a significant global health concern, particularly in the Middle East and North Africa (MENA) region, including Iran [32–34]. Our study provides important insights into the CVD landscape in Hormozgan Province, revealing a prevalence of 7.8% (CI=7.1%−8.4%), with IHD accounting for 6.7% (4.4%−9.0%) and stroke for 2.2% (1.8%−2.5%). Compared to global data, the prevalence observed in our study can be considered moderate. Globally, 1 in 12 individuals live with heart or circulatory diseases, corresponds to an estimated prevalence of around 8.3% [35]. This closely aligns with the prevalence found in our population. These findings reflect alarming trends in morbidity and mortality linked to CVD across the region, driven by rising rates of obesity, HTN, and diabetes [36,37].The demographic characteristics of our study participants indicate a substantial burden of risk factors, including high rates of abdominal obesity (69.6%), HTN (24%), and diabetes (20.6%). Notably, the average FBS of 109mg/dL reveals a concerning trend toward increasing diabetes prevalence in the near future. Furthermore, over half of the participants (57.6%) reported physical inactivity, a situation that is particularly acute among females. This underscores the urgent need for targeted public health interventions

**Table 4. Comparison of CVD prevalence and risk factors by gender and place of residence[1].**

| Variable | Total (N) | Gender | | P value | Residence | | P value[2] |
|---|---|---|---|---|---|---|---|
| | | Male | Female | | Urban | Rural | |
| **Prevalence of CVD[3]** | 489 (7.8) | 223 (8.5) | 266 (7.2) | 0.058 | 262 (8.1) | 227 (7.5) | 0.373 |
| IHD | 422 (6.7) | 197 (7.5) | 225 (6.1) | 0.027 | 225 (6.9) | 197 (6.5) | 0.477 |
| Stroke | 136 (2.2) | 63 (2.4) | 73 (2) | 0.253 | 76 (2.3) | 60 (2) | 0.318 |
| **BMI (kg/m²)** Underweight (<18.5) Normal (18.5–24.9) Overweight (25–29.9) Obese (>30) | 459 (7.3) 2955 (47) 2014 (32) 861 (13.7) | 188 (7.2) 1344 (51.4) 818 (31.3) 263 (10.1) | 271 (7.4) 1611 (43.8) 1196 (32.5) 598 (16.3) | <0.001 | 193 (5.9) 1444 (44.5) 1119 (34.5) 492 (15.1) | 266 (8.7) 1511 (49.7) 895 (29.4) 369 (12.1) | <0.001 |
| **WHR** (Male >0.90, Female >0.85) | 4377 (69.6) | 1631 (62.4) | 2746 (74.7) | <0.001 | 2351 (72.4) | 2026 (66.6) | <0.001 |
| **Physical inactivity** | 3623 (57.6) | 1218 (46.6) | 2405 (65.4) | <0.001 | 2078 (64) | 1545 (50.8) | <0.001 |
| **Smoking** Cigarette Hookah Other Both | 650 (10.3) 215 (33.1) 357 (54.9) 48 (7.4) 30 (4.6) | 387 (14.8) 214 (55.3) 102 (26.4) 43 (11.1) 28 (7.2) | 263 (7.2) 1 (0.4) 255 (97) 5 (1.9) 2 (0.8) | <0.001 <0.001 | 338 (10.4) 125 (37) 182 (53.8) 14 (4.1) 17 (5) | 312 (10.3) 90 (28.8) 175 (56.1) 34 (10.9) 13 (4.2) | <0.001 <0.001 |
| **Diabetes** Prediabetes | 837 (20.6) 1136 (32.1) | 318 (21.7) 449 (35.1) | 519 (20) 687 (30.4) | 0.182 0.004 | 413 (21.1) 569 (30.4) | 424 (20.1) 648 (31.7) | 0.421 0.151 |
| **HTN** Pre-HTN | 1509 (24) 2046 (39.1) | 574 (22) 994 (44.4) | 935 (25.4) 1052 (35.1) | 0.002 <0.001 | 775 (23.9) 1160 (42.2) | 734 (24.1) 886 (35.6) | 0.798 <0.001 |
| **Hypercholesterolemia** | 1624 (40.2) | 580 (39.6) | 1044 (40.6) | 0.561 | 796 (41.0) | 828 (39.5) | 0.301 |
| **Depression** None Mild Moderate Severe | 4533 (72.1) 1013 (16.1) 633 (10.1) 110 (1.7) | 1904 (72.9) 393 (15) 275 (10.5) 41 (1.6) | 2629 (71.5) 620 (16.9) 358 (9.7) 69 (1.9) | 0.150 | 2277 (70.1) 561 (17.3) 368 (11.3) 42 (1.3) | 2256 (74.2) 452 (14.9) 265 (8.7) 68 (2.2) | <0.001 |
| **Anxiety** None Mild Moderate Severe | 5073 (80.7) 795 (12.6) 310 (4.9) 111 (1.8) | 2156 (82.5) 312 (11.9) 109 (4.2) 36 (1.4) | 2917 (79.4) 483 (13.1) 201 (5.5) 75 (2) | 0.006 | 2566 (79) 467 (14.4) 175 (5.4) 40 (1.2) | 2507 (82.4) 328 (10.8) 135 (4.4) 71 (2.4) | <0.001 |

[1]Values are reported as frequency (%).

[2]P values computed by Chi-square.

[3]**CVD** was defined as the presence of IHD or ischemic and hemorrhagic strokes. **Diabetes:** FBS ≥ 126 │medication, **Prediabetes**: FBS 100–126 & no medication, **HTN**: SBP ≥ 140│DBP ≥ 90│medication, **Pre-HTN**: SBP: 130–139.9│DBP:85–89.9 & no medication, **Hypercholesterolemia**: Chol≥ 200│medication.

CVD Cardiovascular Disease, **IHD**: Ischemic Heart Disease, **BMI:** Body Mass Index, **WHR:** Waist-to-Hip Ratio, **HTN:** hypertension, **Pre-HTN:** prehypertension.

designed to encourage more active lifestyles. Our observed treatment success rate for diabetes is aligned with international rates. Systematic reviews and worldwide surveys estimate that the portion of diabetes patients who achieve adequate care varies from approximately 45%, with lower coverage in low- and middle-income countries [38]. For HTN, a worldwide survey of blood pressure control reported an overall success rate of only 13%, far lower than our results. Control rates were worst in lower-middle and lower-income countries (10.8%) compared to 19% in high-income and 15.6% in upper-middle-income countries [39]. several factors are accountable for these low treatment and control rates of HTN, which include limited therapeutic interventions, low healthcare accessibility, low patient education, non-compliance with prescribed medications, and more general socio-economic constraints. Solution of these determinants by targeted interventions is crucial to the enhancement of screening and management protocols of these chronic diseases [40].

 

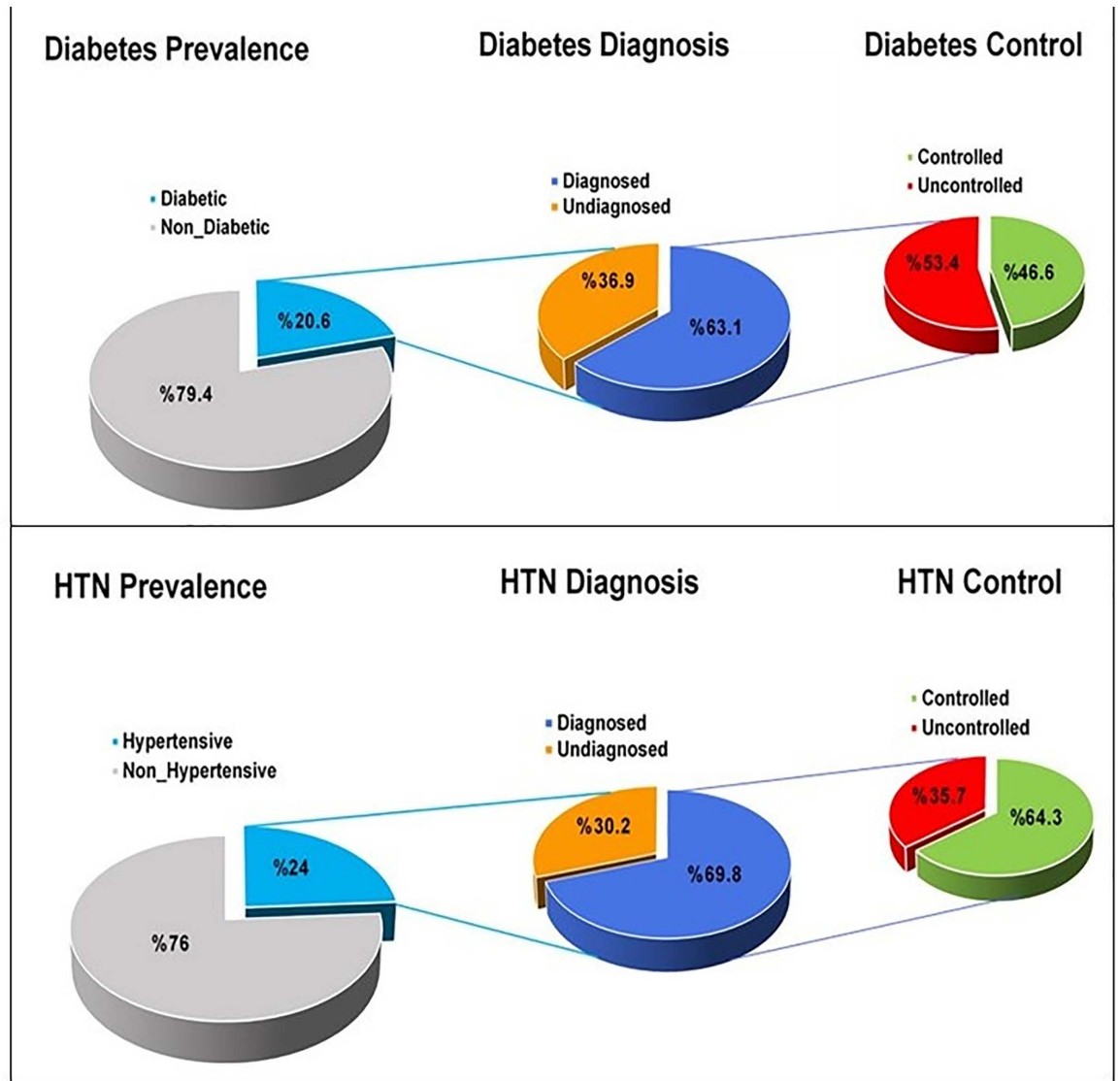

**Fig 2. Distribution of individuals by prevalence, diagnosis, and control of diabetes and HTN.**

Our logistic regression analysis showed that increased age, increased BMI, hypercholesterolemia, diabetes mellitus, HTN, and severe anxiety were significantly linked to an increased prevalence of CVD. Of those, HTN showed the strongest association and demonstrated very good concordance with findings from global research. HTN has been established globally as the most potent modifiable risk factor for CVD incidence and death [41,42]. In addition, more recent data indicate that psychiatric disorders such as anxiety can be deleterious to cardio metabolic well-being, particularly when compounded by comorbid obesity and other metabolic derangements [43]. Our findings support previous research indicating that increased age is associated with a higher incidence of CVD, likely due to vascular aging, endothelial dysfunction, and cumulative exposure to risk factors [44,45]. While age is closely linked to an increased risk of CVD, this association is primarily influenced by complex factors such as comorbidities, lifestyle, gender, genetics, and biological aging processes [44,46], suggesting that age may serve as a surrogate marker rather than an independent causal factor. Similarly,

**Table 5. Univariate logistic regression analysis for the association of characteristics and metabolic variables with CVD.**

| Variable | COR | 95% CI | P value |
|---|---|---|---|
| **Age** (years) | 1.042 | (1.036, 1.048) | <0.001 |
| **Sex** (Female) | 0.836 | (0.694, 1.006) | 0.058 |
| **Marital status** | | | |
| Single | Ref | – | – |
| Married | 2.249 | (1.521, 3.326) | <0.001 |
| Divorce/Widow | 3.891 | (2.478, 6.110) | <0.001 |
| **Education** | | | |
| Illiteracy (<1) year | Ref | – | – |
| Primary school (1–5) | 0.784 | (0.586, 1.049) | 0.101 |
| Secondary school (6–8) | 0.366 | (0.276, 0.486) | <0.001 |
| High school (9–12) | 0.321 | (0.242, 0.425) | <0.001 |
| Bachelor's degree (13–16) | 0.277 | (0.193, 0.398) | <0.001 |
| Masters/PhD degree (>16) | 0.270 | (0.097, 0.754) | 0.012 |
| **Occupation** | | | |
| Employed | Ref | – | – |
| Unemployed | 1.648 | (1.167, 2.327) | 0.005 |
| Housewife | 1.188 | (0.960, 1.469) | 0.112 |
| Retired | 3.116 | (2.211, 4.392) | <0.001 |
| **Wealth index** | | | |
| Quintile 1 (Poorest) | Ref | – | – |
| Quintile 2 | 1.002 | (0.765, 1.311) | 0.990 |
| Quintile 3 | 0.640 | (0.474, 0.863) | 0.003 |
| Quintile 4 | 0.771 | (0.580, 1.026) | 0.074 |
| Quintile 5 (Richest) | 0.718 | (0.537, 0.960) | 0.025 |
| **Depression** | | | |
| None | Ref | – | – |
| Mild | 1.370 | (1.074, 1.748) | 0.011 |
| Moderate | 1.816 | (1.388, 2.376) | <0.001 |
| Severe | 2.166 | (1.241, 3.779) | 0.007 |
| **Anxiety** | | | |
| None | Ref | – | – |
| Mild | 1.372 | (1.058, 1.778) | 0.017 |
| Moderate | 1.494 | (1.020, 2.186) | 0.039 |
| Severe | 2.511 | (1.499, 4.207) | <0.001 |
| **Physical inactivity** (Yes) | 1.228 | (1.016, 1.485) | 0.034 |
| **Smoking** (Yes) | 1.038 | (0.764, 1.411) | 0.812 |
| **BMI** (kg/m$^2$) | 1.031 | (1.012, 1.051) | 0.001 |
| **WHR** | 2.498 | (1.157, 5.395) | 0.020 |
| **Diabetes** (Yes) | 2.470 | (1.957, 3.117) | <0.001 |
| **HTN** (Yes) | 3.441 | (2.852, 4.153) | <0.001 |
| **Hypercholesterolemia** (Yes) | 1.936 | (1.553, 2.413) | <0.001 |

**COR**: Crude Odds Ratios, **CVD**: Cardiovascular Disease, **BMI**: Body Mass Index, **WHR**: Waist to Hip Ratio, **HTN**: Hypertension

the positive relationship between higher BMI and CVD corroborates previous studies that identify obesity as a significant risk factor for CVD through mechanisms like inflammation, insulin resistance, and dyslipidemia [47,48]. Additionally, we found that hypercholesterolemia is linked to CVD, confirming evidence that high cholesterol levels lead to atherosclerosis and related cardiovascular events [49,50]. Moreover, there is a significant link between diabetes and CVD, consistent with studies that show diabetes increases the risk of cardiovascular complications [51,52]. Notably, the lower CVD risk observed in females compared to males in this study aligns with findings from numerous previous studies [53].

**Table 6. Multivariable logistic regression analysis for the association of characteristics and metabolic variables with CVD.**

| Variable | AOR | 95% CI | P value |
|---|---|---|---|
| **Age** (years) | 1.033 | (1.024, 1.043) | <0.001 |
| **BMI** (kg/m²) | 1.029 | (1.004, 1.055) | 0.021 |
| **Sex** (Female) | 0.676 | (0.533, 0.857) | 0.001 |
| **Hypercholesterolemia** (Yes) | 1.308 | (1.031, 1.661) | 0.027 |
| **HTN** (Yes) | 2.026 | (1.550, 2.649) | <0.001 |
| **Diabetes** (Yes) | 1.313 | (1.006, 1.714) | 0.045 |
| **Anxiety** | | | |
| Normal | Ref | – | – |
| Mild | 1.188 | (0.853, 1.653) | 0.309 |
| Moderate | 1.083 | (0.673, 1.743) | 0.743 |
| Severe | 2.392 | (1.303, 4.390) | 0.005 |

**AOR**: Adjusted Odds Ratios, **CVD**: Cardiovascular Disease, **BMI**: Body Mass Index, **HTN**: Hypertension.

Overall, the findings of this study reinforce existing research and underscore the need to address traditional and emerging risk factors—such as age, BMI, lipid profiles, diabetes, hypertension, mental health, and gender differences—in comprehensive strategies for preventing and managing CVD. While findings corroborate the established role of traditional risk factors, they also highlight a critical gap in the contemporary evidence: a lack of understanding regarding the complex interplay and relative weighting of these factors in driving CVD risk. Future research should focus on developing integrated risk models that account for synergistic effects (e.g., how diabetes and hypercholesterolemia interact differently by gender or age). This will be essential for advancing personalized, rather than generic, prevention strategies.

Whereas smoking has been widely established as a significant risk factor for CVD, our study was not able to demonstrate a statistically significant relationship (OR = 1.038, p = 0.812). Our result is opposite to a number of previous studies and meta-analyses that have identified excess cardiovascular risk of 44–55% in smokers. There are a number of reasons for this inconsistency, including biases in self-reporting, regional or cultural variations in smoking behavior, and differences in study design [54–57]. These findings, therefore, need to be interpreted with caution, and further studies are necessary to clarify the role of smoking in specific populations. It must be stated that the cross-sectional nature of this research does not allow us to make causal inferences about associations between CVD and risk factors.

Given the multifactorial nature of cardiovascular disease risk, effective control requires a multifaceted approach. The WHO recommends medication, a balanced diet, and physical activity as key measures to reduce the global CVD burden, a framework supported by our findings. Additionally, the observed association between severe anxiety and CVD highlights the importance of integrating mental health support into prevention and care strategies.

Based on these findings, we propose the following actions:

• Developing and implementing comprehensive public health strategies targeting the prevention and control of HTN, diabetes.

• Strengthening healthcare systems to improve screening, diagnosis, and management of CVD risk factors.

• Incorporating mental health screening and interventions into routine CVD prevention protocols.

To maximize impact, policymakers and healthcare professionals should collaborate to address shared risk factors, thereby reducing CVD burden and improving cardiovascular health outcomes in Hormozgan Province and similar regions.

## Strengths and limitations

In order to enhance the validity and generalizability of the study, participants were recruited from both genders, with a variety of educational backgrounds, occupations, and socioeconomic statuses. The study's strengths included valuable infrastructure, investigation of CVD risk factors, biobank creation, and extensive baseline data collection.

A key limitation of this study is the cross-sectional design, which hinders the ability to infer temporal or causal relationships between CVD and risk factors. Additionally, cultural restrictions on data collection, lower participation among housebound individuals, and potential information bias from self-reported questionnaires may have influenced the results.

The causal relationship between these risk factors and CVD is well established, and this study does not primarily aim to explore that relationship. Instead, updating our understanding of current trends and the prevailing prevalence of these factors enables us to make more informed decisions and take more effective actions than in the past.

## Conclusion

The prevalence of CVD was 7.8%, which can be considered moderate when compared to other populations. Key factors linked to CVD included advanced age, increased BMI, high cholesterol levels, diabetes, hypertension, and severe anxiety. Notably, females in this study were found to have a lower risk of developing CVD compared to males. This study highlights the increasing burden of CVD in Hormozgan Province and emphasizes the significant impact of modifiable lifestyle factors such as obesity, HTN, diabetes, physical inactivity, and severe anxiety. These findings indicate a need for improved healthcare access, preventive screening, and targeted health education regarding these risk factors.

## Supporting information

**S1 File. Data.**
(XLSX)

## Acknowledgments

We sincerely thank Dr. Gholamali Javedan, the chancellor of Hormozgan University of Medical Sciences, for his steadfast support. Additionally, we would like to express our appreciation to the Deputy of Research and Technology at Hormozgan University of Medical Sciences for his essential guidance during this project. Our appreciation goes to the Non-Communicable Diseases Unit of the Health Deputy for their crucial role in advancing our research objectives. Finally, we thank the dedicated staff at local health centers and all the participants for their time and insights, which were essential to the success of this study.

## Author contributions

**Conceptualization:** Hossein Farshidi, Marzieh Nikparvar, Shokrollah Mohseni, Abdoljabbar Zakeri.

**Data curation:** Hossein Farshidi, Marzieh Nikparvar, Abdoljabbar Zakeri, Shahin Abbaszadeh.

**Formal analysis:** Shokrollah Mohseni, Farideh Dastsouz, Abdoljabbar Zakeri.

**Funding acquisition:** Hossein Farshidi, Marzieh Nikparvar, Abdoljabbar Zakeri.

**Investigation:** Hossein Farshidi, Marzieh Nikparvar, Shokrollah Mohseni, Ebrahim Eftekhar, Farideh Dastsouz, Abdoljabbar Zakeri, Mohammad Mohammadi, Farkhondeh Razmpour, Farid Khorrami, Shahin Abbaszadeh.

**Methodology:** Hossein Farshidi, Marzieh Nikparvar, Shokrollah Mohseni, Ebrahim Eftekhar, Abdoljabbar Zakeri, Mohammad Mohammadi, Farkhondeh Razmpour, Farid Khorrami, Soghra Fallahi, Shahin Abbaszadeh.

**Project administration:** Marzieh Nikparvar, Abdoljabbar Zakeri, Shahin Abbaszadeh.

**Software:** Hossein Farshidi, Marzieh Nikparvar, Shokrollah Mohseni, Ebrahim Eftekhar, Farideh Dastsouz, Mohammad Mohammadi, Farid Khorrami, Masoumeh Mahmoodi, Shahin Abbaszadeh.

**Supervision:** Shokrollah Mohseni, Ebrahim Eftekhar, Farideh Dastsouz, Soghra Fallahi, Atefeh Ghareghani, Najmeh Shanbehzadeh, Shahin Abbaszadeh.

**Validation:** Shokrollah Mohseni, Shahin Abbaszadeh.

**Visualization:** Shahin Abbaszadeh.

**Writing – original draft:** Shokrollah Mohseni, Ebrahim Eftekhar, Farideh Dastsouz, Masoumeh Mahmoodi, Nahid Shahabi, Soghra Fallahi, Najmeh Shanbehzadeh, Shahin Abbaszadeh.

**Writing – review & editing:** Shokrollah Mohseni, Ebrahim Eftekhar, Farideh Dastsouz, Masoumeh Mahmoodi, Nahid Shahabi, Soghra Fallahi, Najmeh Shanbehzadeh, Shahin Abbaszadeh.

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
