## [Decision Letter · Decision Letter 0]

27 Jun 2025

Dear Dr. Abbaszadeh,

Thank you for submitting your manuscript to PLOS ONE. After careful consideration, we feel that it has merit but does not fully meet PLOS ONE’s publication criteria as it currently stands. Therefore, we invite you to submit a revised version of the manuscript that addresses the points raised during the review process.

We look forward to receiving your revised manuscript.

Kind regards,

Amin Mansoori

Academic Editor

PLOS ONE

Journal Requirements: 

Additional Editor Comments:

Dear Authors,

Thank you for submitting your manuscript to our journal. We have now received feedback from four expert reviewers, who acknowledge that your work presents a novel idea and has the potential to be publishable. However, they have also raised several serious concerns regarding the conceptual framework, grammatical issues, and the presentation and discussion of your findings.

After careful consideration of the reviewers' comments, I have decided to offer you the opportunity to revise your manuscript. I strongly encourage you to address all concerns thoroughly, revising the text with great care. Please provide a detailed, point-by-point response to each of the reviewers' comments, along with the necessary modifications in the manuscript.

We look forward to receiving your revised version.

Reviewers' comments:

Reviewer's Responses to Questions

**Comments to the Author**

1. Is the manuscript technically sound, and do the data support the conclusions?

Reviewer #1: Yes

Reviewer #2: Partly

Reviewer #3: Yes

Reviewer #4: Yes

2. Has the statistical analysis been performed appropriately and rigorously?

Reviewer #1: Yes

Reviewer #2: Yes

Reviewer #3: No

Reviewer #4: Yes

3. Have the authors made all data underlying the findings in their manuscript fully available?

Reviewer #1: Yes

Reviewer #2: Yes

Reviewer #3: Yes

Reviewer #4: No

4. Is the manuscript presented in an intelligible fashion and written in standard English?

Reviewer #1: No

Reviewer #2: Yes

Reviewer #3: No

Reviewer #4: Yes

Reviewer #1: The study was conducted among good numbers of samples with reasonable rate of response rate (Though little low for face-to-face but okay for online survey). Reliable tools (3 item UCLA loneliness Scale, and SAS_SV for PSU were utilized which are with good internal consistency. Some suggestions are:

1. Avoid sub-headings in introduction section.

2. Line 55, need to revise (Grammatical and other issue (i.e., stagma) need to be verified and redo.

Line 135, No need to keep question here, just attach as annexure.

3. Line 193: Why total is kept in first row? I suggest to keep total in last row.

4. Line 210: There is no need to keep each tools' internal consistency score in description of results.

5. Chronbatch alpha may be calculated again if the tool was modified in local language.

Reviewer #2: 1. In abstract include number of participants not sample size 6504...6298

2. In abstract result part: prevalence of abdominal obesity, DM , hypertension and success rate of treatment … Not needed; it will not go with the objectives of the study. Which BMI category, age, and sex were associated with CVD?

3. Line 111: remove Participants aged 20 and older were recruited from design and geography… it was already in participant/population

4. Line 114/5: The study received approval from the Ethics Committee of Hormozgan University of Medical Sciences (IR.HUMS.REC.1401.080). … Include it in the ethical part.

5. Line 231 definition…>make it operational definition

6. Line 234/35: … Medical documents were scanned and reviewed by a cardiologist to confirm the definitive diagnosis. This is for IHD. How ischemic stroke or hemorrhagic stroke was diagnosed was not explained.

7. Line 260...: At what p-value was statistical significance for all the independent variables in the final model declared? because you described that Variables that exhibited a p-value of less than 0.05 in the univariate analysis were

included in the multivariable logistic regression model? How was the strength of association between independent and dependent variables assessed? Explain whether you use COR/AOR with a % confidence interval.

8. Line 358: prevalence of 7.8%....add confidence interval of prevalence

9. Lines 368-375 There is no need to discuss smoking and hookah use, as there is no finding in this study.

10. Discussion and recommendation are mixed

11. Discussion was not elaborated by comparing and contrasting findings with previous studies/literature

12. Conclusion and recommendation were mixed. Conclusion was not narrating the research finding… focus on your findings

13. No recommendation was given on isolated topic

Reviewer #3: Dear Editor,

I hope this message finds you well. I have completed my review of the manuscript entitled “Exploring Cardiovascular Disease Prevalence and Contributing Factors among Adults in Southern Iran, a Cross-Sectional Survey: Rationale, Design, and Primary Results” with manuscript Number: PONE-D-25-24170. Below, you will find my detailed comments and suggested revisions for your consideration.

Thank you for the opportunity to contribute to the peer review process.

Sincerely,

Comment 1:

Please revise the use of abbreviations throughout the manuscript. Ensure that each abbreviation is spelled out in full at its first occurrence, followed by the abbreviated form in parentheses.

Comment 2:

The manuscript contains several grammatical errors that affect readability and clarity. Additionally, ensure there is a space between the final word of a sentence and the subsequent reference citation. Please revise the entire manuscript carefully to correct grammatical inaccuracies, improve clarity, and ensure consistent formatting throughout.

Comment 3:

While the authors describe the use of cluster and stratified random sampling, the methodology lacks sufficient detail to assess its rigor. Please clarify:

• Cluster Selection: How were clusters defined (e.g., geographic units, institutions)? What criteria were used for their selection?

• Stratification: On what basis were strata established (e.g., socioeconomic status, disease prevalence)? How were sampling proportions determined?

• Urban/Rural Representation: Was the allocation proportional to population distribution? If not, justify the approach.

A flow diagram or visual summary of the sampling strategy would greatly enhance reproducibility and transparency.

Comment 4:

Although the use of WHO STEPS and GPAQ tools is noted, there is no mention of whether the questionnaire was pilot tested or validated in the local population. Please specify whether the adapted version underwent pretesting, and if so, how reliability and validity were assessed. This is crucial given cultural and linguistic differences that may affect data quality.

Comment 5:

While the use of SPSS and basic statistical tests is appropriate, the manuscript requires further clarification regarding the model-building strategy for the multivariable logistic regression. Specifically, please address the following:

• Variable Selection: How were confounding variables identified and selected for inclusion?

• Multicollinearity: Was multicollinearity assessed among predictors? If so, what measures were used (e.g., variance inflation factors, correlation matrices)?

• Inclusion/Exclusion Criteria: What statistical or theoretical criteria determined the final variables retained in the model (e.g., p-value thresholds, likelihood ratio tests)?

• Interaction Terms: Were potential interaction terms explored? If yes, how were they selected and evaluated?

Providing these details will enhance the reproducibility and methodological rigor of the analysis.

Comment 6:

Figures 1 and 2 are not properly indexed or cited within the main body of the manuscript. Currently, only the figure legends appear in the text, without any corresponding references or placements indicating where the figures should be consulted. Please ensure that both figures are explicitly referenced and appropriately positioned within the text to guide readers and enhance the clarity of the study’s methodology and results.

Comment 7:

The Discussion section does not adequately contextualize the study's findings in relation to previous research. A critical component of a well-structured discussion is the comparison and contrast of current results with those reported in existing literature. At present, this section lacks sufficient integration of relevant studies and does not effectively highlight consistencies or discrepancies with past findings. I recommend revising the Discussion to include a more comprehensive and analytical comparison with previous studies in your area, thereby strengthening the interpretation and significance of the results.

Reviewer #4: No. the authors mention data restrictions. A thorough language revision by a native English speaker is recommended to enhance readability.

This cross-sectional study by Farshdi et al investigates the prevalence, risk factors, and management of cardiovascular disease (CVD) in Hormozgan Province, Iran. The study includes data from over 6,500 males and females participants aged 20 and older using the WHO steps methodology with a high response rate (96.6%), increasing the validity and generalizability of findings to the local population. The authors highlight key sociodemographic, metabolic, behavioural, and psychological risk factors. While the study is valuable and timely, particularly in the context of growing noncommunicable disease burdens in the MENA region. However, some methodological and reporting issues need clarification to better understand the underlying the clinical relevance of these findings.

Major comments

The results of the analyses are interesting if somewhat predictable. Since it is not a longitudinal study, as a cross-sectional study, we can not interpret the association between risk factors and CVD. Authors need to clearly state this in the discussion and limitation section.

Table 5 shows the regression analysis of the variables such as depression, anxiety and son, but the definition and methods for the measurement of these variables are missing. Detailed methodology on how these variables were measured need to added with cut-offs or scoring system.

Univariate analysis show depression and anxiety are associated with CVD, however only severe anxiety remains significant in multivariate analysis. Again, authors need to explain how severity was categorized.

Table 5, smoking shows no association with CVD (OR = 1.038, p = 0.812), that contradicts with the literature or it could be likely self-reported bias and might not be fully reliable.

Minor Comments

Minor grammatical errors should be addressed to improve readability. All legends should explain all terms and abbreviations used.

**Do you want your identity to be public for this peer review?** For information about this choice, including consent withdrawal, please see our Privacy Policy

Reviewer #1: **Yes: ** Dr. Kishor Adhikari

Reviewer #2: **Yes: ** DEBELA EJETA GEDEFA

Reviewer #3: No

Reviewer #4: No

---

## [Author Response · Author response to Decision Letter 1]

11 Aug 2025

Response to reviewers

Dear Editor and Reviewers,

We sincerely thank you for the time and effort invested in reviewing our manuscript. We greatly appreciate the insightful comments and constructive feedback provided, which have helped us to improve the quality of our case report. Below, we address each of the reviewer’s points in detail.

Reviewer 1:

The study was conducted among good numbers of samples with reasonable rate of response rate (Though little low for face-to-face but okay for online survey). Reliable tools (3 item UCLA loneliness Scale, and SAS_SV for PSU were utilized which are with good internal consistency. Some suggestions are:

1. Avoid sub-headings in introduction section.

Author response: We sincerely appreciate the reviewer’s thoughtful comment. We confirm that no sub-headings have been included in the Introduction section of the manuscript.

2. Line 55, need to revise (Grammatical and other issue (i.e., stagma) need to be verified and redo.

Author response: We thank the reviewer for highlighting this potential issue. Based on the submitted PDF version, there is no text at line 55. To ensure we address your concern accurately, we revised the text through out to check grammar errors. Please let us know if we need to improve any other specific section.

3. Line 135, No need to keep question here, just attach as annexure.

Author response: We sincerely appreciate the reviewer’s suggestion. However, we would like to respectfully clarify that the questionnaire is not included in the main text of our submitted manuscript. Moreover, according to the submitted PDF version, line 135 does not contain any reference to the questionnaire. We would be grateful if the reviewer could kindly provide further guidance or specify the exact section or text where the questionnaire is mentioned, so that we can address the concern appropriately.

4. Line 193: Why total is kept in first row? I suggest to keep total in last row.

Author response: We thank the reviewer for this helpful suggestion. However, since the total population data are a key reference point in our analysis, we deliberately placed the “Total” row at the top of the table to highlight the overview of the data for readers.

5. Line 210: There is no need to keep each tool's internal consistency score in description of results.

Author response: We are grateful for the reviewer’s attention to detail. According to the PDF we submitted, there is no mention of internal consistency scores at line 210. If this comment refers to another section, could you please advise us so we can make the appropriate revision? We are happy to address this promptly with your guidance.

6. Chronbach alpha may be calculated again if the tool was modified in local language.

Author response: We truly appreciate this important point. The questionnaires used in our study had already undergone validation and reliability testing for the Iranian population in Persian. No further modifications were made in this study; therefore, recalculation of Cronbach’s alpha was not necessary.

We highly value your constructive feedback, which helps improve the clarity and quality of our manuscript. Should you require additional information or clarification on any point—especially regarding lines 55 and 210—please let us know.

Reviewer 2:

1. In abstract include number of participants not sample size 6504...6298

Author Response: Thank you for this suggestion. We have updated the abstract to clearly state the exact number of participants included in the study.

Page 2, lines 29

2. In abstract result part: prevalence of abdominal obesity, DM, hypertension and success rate of treatment … Not needed; it will not go with the objectives of the study. Which BMI category, age, and sex were associated with CVD?

Author Response: We sincerely thank the reviewer for their valuable comments, which have helped us improve the clarity and focus of our manuscript. In response, we have revised the Results section of the abstract by removing the prevalence data related to abdominal obesity, diabetes mellitus, hypertension, and treatment success rates. Instead, we have emphasized the associations between BMI, age, and sex with cardiovascular disease (CVD). however, both age and body mass index were analyzed as continuous variables. Our findings indicated that higher age and BMI are independently associated with an increased risk of CVD. Additionally, our analysis showed that females have a 33% lower risk of developing CVD compared to males (OR: 0.67; 95% CI: 0.53–0.85).

Page 2, lines 40-44, Page 3, lines 47.

The revised abstract results are as follows: Results: A total of 6,298 participants were included in the study, with a mean age of 45.63 ± 15.04 years. Among them, 58.5% were female, and 51.6% lived in urban areas. Key findings revealed that 43.9% of the population were identified as overweight or obese, with a significant prevalence of abdominal obesity at 69.6%, particularly among females. The prevalence of CVD in the population was 7.8%, with IHD affecting 6.7% of individuals and stroke affecting 2.2%. Multivariable logistic regression identified increased age (AOR: 1.03; 95% CI: 1.02–1.04), higher body mass index (BMI) (AOR: 1.02; 95% CI: 1.00–1.05), hypercholesterolemia (AOR: 1.30; 95% CI: 1.03–1.66), hypertension (HTN) (AOR: 2.02; 95% CI: 1.55–2.64), diabetes (AOR: 1.31; 95% CI: 1.00–1.71), and severe anxiety (AOR: 2.39; 95% CI: 1.30–4.39) as significant risk factors for CVD. Women had a 33% lower risk of having CVD compared to men (AOR: 0.67; 95% CI: 0.53–0.85).

3. Line 111: remove "Participants aged 20 and older were recruited from design and geography…" as it was already in participant/population section

Author Response: Thank you for your constructive feedback, The redundant sentence has been removed for clarity and conciseness.

4. Line 114/5: The study received approval from the Ethics Committee of Hormozgan University of Medical Sciences (IR.HUMS.REC.1401.080). Include it in the ethical part.

Author Response: We appreciate the reviewer’s comments. In accordance with your suggestion, the ethical approval information has been moved to the "Ethics Approval and Consent to Participate" section, after the Data Availability statement.

Page 33, lines 549-551

5. Line 231 definition…>make it operational definition

Author Response: Thank you for your careful reading of our manuscript and for your helpful comments, Definitions have been revised and clarified as operational definitions.

6. Line 234/35: … Medical documents were scanned and reviewed by a cardiologist to confirm the definitive diagnosis. For IHD. How ischemic stroke or hemorrhagic stroke was diagnosed was not explained.

Author Response: We are grateful for the reviewer’s thorough review and helpful suggestions, Additional details about the diagnostic criteria and confirmation process for ischemic and hemorrhagic stroke have been included to clarify this point.

Page 14, lines 274-276

7. Line 260: At what p-value was statistical significance for all independent variables in the final model declared? How was the strength of association between independent and dependent variables assessed? Explain whether you used COR/AOR with a confidence interval.

Author Response: Thank you for your constructive feedback,We have specified that a p-value < 0.05 was used to define statistical significance. Additionally, we clarified that both Crude Odds Ratios (COR) and Adjusted Odds Ratios (AOR) with 95% confidence intervals were calculated to assess the strength of associations.

Page 15, lines 299-306

The revised Data analysis tools and methods section is as follows: All statistical analyses were conducted using SPSS version 26. Quantitative variables were described using means and standard deviations, while categorical and qualitative variables were represented by frequency counts and percentages. The Chi-square test was employed to compare the general characteristics of subjects between the two groups for qualitative variables. The Kolmogorov-Smirnov test was used to evaluate the normality of the data distribution. The Mann-Whitney test was applied to quantitative variables that did not exhibit a normal distribution. Univariate and multivariable logistic regressions were conducted to identify associations between various characteristics and metabolic variables with CVD. Variables that yielded a p-value less than 0.2 in the univariate analysis were included in the multivariable logistic regression model. In the multivariable model, we systematically removed variables with the highest p-values until all remaining variables had p-values below 0.05. Both crude odds ratios (COR) and adjusted odds ratios (AOR), along with their 95% confidence intervals (CIs), were calculated to quantify the strength of associations between independent variables and CVD. A p-value less than 0.05 was considered statistically significant.

8. Line 358: prevalence of 7.8%....add confidence interval of prevalence

Author Response: The confidence interval has been added to the prevalence estimate as requested.

Page 26, lines 410

9. Lines 368-375: There is no need to discuss smoking and hookah use, as there is no finding in this study.

Author response: Thanks for your comment, The discussion related to smoking and hookah use has been removed.

10. Discussion and recommendation are mixed

Author Response: Thank you for your feedback on our manuscript. We made the changes accordingly. The discussion and recommendations have been separated into distinct sections for clarity.

Page 27-29, lines 434-479

11. Discussion was not elaborated by comparing and contrasting findings with previous studies/literature

Author Response: Thank you for your feedback on our manuscript. We appreciate your comments, which will help improve our paper. As suggested by the esteemed reviewer, we have compared and discussed our study’s results in relation to findings from several relevant articles.

Page 27-28, lines 434-460

12. Conclusion and recommendation were mixed. Conclusion was not narrating the research finding… focus on your findings

Author Response: Thank you for your comments ,The conclusion has been rewritten to focus clearly on the study’s key findings, with recommendations presented separately.

Page 31-32, lines 529-533

13. No recommendation was given on isolated topic

Author Response: Specific recommendations related to isolated topics have now been added to the recommendation section.

Page 29, lines 464-479

Reviewer 3:

Reviewer #3: Dear Editor,

I hope this message finds you well. I have completed my review of the manuscript entitled “Exploring Cardiovascular Disease Prevalence and Contributing Factors among Adults in Southern Iran, a Cross-Sectional Survey: Rationale, Design, and Primary Results” with manuscript Number: PONE-D-25-24170. Below, you will find my detailed comments and suggested revisions for your consideration.

Thank you for the opportunity to contribute to the peer review process.

Sincerely,

Author response: Thank you very much for your thorough and constructive review of our manuscript. We appreciate the time and expertise you have invested in helping us improve our work. Below we address each of your comments in detail.

Comment 1:

Please revise the use of abbreviations throughout the manuscript. Ensure that each abbreviation is spelled out in full at its first occurrence, followed by the abbreviated form in parentheses.

Author response: We have carefully reviewed the manuscript and ensured that each abbreviation is spelled out in full at its first occurrence, followed by the abbreviation in parentheses. Subsequent use of abbreviations is consistent throughout the text.

Comment 2:

The manuscript contains several grammatical errors that affect readability and clarity. Additionally, ensure there is a space between the final word of a sentence and the subsequent reference citation. Please revise the entire manuscript carefully to correct grammatical inaccuracies, improve clarity, and ensure consistent formatting throughout.

Author response: We have thoroughly revised the manuscript to correct grammatical errors and improve overall readability and clarity. Additionally, we reviewed all reference citations to confirm appropriate spacing between the final word and the citation.

Comment 3:

While the authors describe the use of cluster and stratified random sampling, the methodology lacks sufficient detail to assess its rigor. Please clarify:

Author response: We greatly appreciate the reviewer’s pertinent observations regarding our sampling methodology. In response, we have revised and expanded the Methods section to provide greater clarity and detail.

• Cluster Selection: How were clusters defined (e.g., geographic units, institutions)? What criteria were used for their selection?

Author response: Clusters were defined as individual health service units (comprehensive health centers or health posts in urban areas; health service units in rural areas), and selected through simple random sampling from an updated registry.

Page 8, lines 147-158

• Stratification: On what basis were strata established (e.g., socioeconomic status, disease prevalence)? How were sampling proportions determined?

Author response: Stratification was conducted at two levels: first by county (geographic stratum), and then by urban/rural residence within each county, with sample allocation in each stage determined proportionally to the size of the adult population.

Page 7-8, lines 144-168

• Urban/Rural Representation: Was the allocation proportional to population distribution? If not, justify the approach.

Author response: Participant allocation between urban and rural areas within each county was strictly proportional to their respective population sizes.

Household selection within clusters was randomized and strictly followed a protocol to mitigate potential bias from non-responders.

Page 7-8, lines 144-168

A flow diagram or visual summary of the sampling strategy would greatly enhance reproducibility and transparency.

Author response: As recommended, a summary table outlining the complete sampling strategy has been included (see Table 1) to facilitate reproducibility and transparency. Page 9, lines 170

Comment 4:

Although the use of WHO STEPS and GPAQ tools is noted, there is no mention of whether the questionnaire was pilot tested or validated in the local population. Please specify whether the adapted version underwent pretesting, and if so, how reliability and validity were assessed. This is crucial given cultural and linguistic differences that may affect data quality.

Author response: We thank the reviewer for highlighting the importance of questionnaire validity and reliability in the study population. In response, we have clarified in the revised manuscript that The validity and reliability of the WHO STEPS instruments and other questionnaires, have been thoroughly evaluated in the Iranian context. All questionnaires employed in this study have previously been utilized in multiple rounds of the national Stepwise Approach to Noncommunicable Disease Risk Factor Surveillance (WHO STEPS) surveys in Iran, specifically in 2005, 2006, 2007, 2008, 2009, 2011, and 2016. Relevant references for these validation and reliability studies are provided in the Methods section of the manuscript to ensure transparency and facilitate further review.

Page 10, lines 201-203

Comment 5:

While the use of SPSS and basic statistical tests is appropriate, the manuscript requires further clarification regarding the model-building strategy for the multivariable logistic regression. Specifically, please address the following:

• Variable Selection: How were confounding variables identified and selected for inclusion?

• Multicollinearity: Was multicollinearity assessed among predictors? If so, what measures were used (e.g., variance inflation factors, correlation matrices)?

• Inclusion/Exclusion Criteria: What statistical or theoretical criteria determined the final variables retained in the model (e.g., p-value thresholds, likelihood ratio tests)?

• Interaction Terms: Were potential interaction terms explored? If yes, how were they selected a

---

## [Decision Letter · Decision Letter 1]

1 Sep 2025

Dear Dr. Abbaszadeh,

Thank you for submitting your manuscript to PLOS ONE. After careful consideration, we feel that it has merit but does not fully meet PLOS ONE’s publication criteria as it currently stands. Therefore, we invite you to submit a revised version of the manuscript that addresses the points raised during the review process.

We look forward to receiving your revised manuscript.

Kind regards,

Amin Mansoori

Academic Editor

PLOS ONE

Journal Requirements:

Additional Editor Comments:

Dear Author,

Thank you for submitting your revised manuscript. Upon evaluation, I find that the paper has improved significantly. However, before we can proceed with acceptance, one of the reviewers has provided some constructive comments to further enhance the manuscript. I encourage you to address these suggestions carefully and submit an updated version for our final consideration. We appreciate your efforts and look forward to receiving your revised manuscript.

Best regards,

Amin Mansoori

Reviewers' comments:

Reviewer's Responses to Questions

**Comments to the Author**

Reviewer #2: (No Response)

Reviewer #3: (No Response)

Reviewer #4: All comments have been addressed

2. Is the manuscript technically sound, and do the data support the conclusions?

Reviewer #2: Yes

Reviewer #3: (No Response)

Reviewer #4: Yes

3. Has the statistical analysis been performed appropriately and rigorously?

Reviewer #2: Yes

Reviewer #3: (No Response)

Reviewer #4: N/A

4. Have the authors made all data underlying the findings in their manuscript fully available?

Reviewer #2: Yes

Reviewer #3: (No Response)

Reviewer #4: Yes

5. Is the manuscript presented in an intelligible fashion and written in standard English?

Reviewer #2: Yes

Reviewer #3: (No Response)

Reviewer #4: Yes

Reviewer #2: 1. Line 408-410: Our study provides important insights into the CVD landscape in Hormozgan Province, revealing a prevalence of 7.8% (CI=0.071-0.084), with IHD accounting for 6.7% (0.044-0.09) and stroke for 2.2% (0.018-0.025)

1.1.Confidence intervals (CI) should be reported in the same unit as the main value.

If you report prevalence as a percentage (e.g., 7.8%), then the CI should also be in percentage (CI: 7.1%–8.4%).

1.2. Discuss this finding(prevalence of CVD) by comparing and contrasting prevalence of CVD from other parts of the world. This helps the reader whether prevalence of CVD in the region is high or low compared to other parts of world.

2. Discussion line 445-451

Our logistic regression analysis …..increased age, increased BMI, hypercholesterolemia, diabetes mellitus, HTN, and severe anxiety….include female sex

Hypertension and severe anxiety was discussed but increased age, increased BMI, hypercholesterolemia, diabetes mellitus and sex should be also discussed by comparing and contrasting with previous studies(whether your finding is similar or different )

3. Line 463-465 and 471-472:Our study further emphasizes tailoring interventions, particularly to increase physical activity among women and urban populations….no finding in your study that suggest physical activity in women and urban population associated to CVD. Remove it

4. Line 469-70: even though prevalence of obesity is high, no finding in your study that suggest obesity associated factor to CVD: better to remove it.

5. Conclusion

Focus on your objectives

example like this. The prevalence of CVD was 7.8%(is it high/low comparing…). Factors significantly associated with CVD included older age, higher BMI, hypercholesterolemia, diabetes mellitus, hypertension, and severe anxiety. Interestingly, female sex appeared to be protective against CVD compared to male sex.

Reviewer #3: (No Response)

Reviewer #4: The authors have addressed all the comments and suggestions.

The manuscript has been significantly improved.

**Do you want your identity to be public for this peer review?** For information about this choice, including consent withdrawal, please see our Privacy Policy

Reviewer #2: **Yes: ** DEBELA EJETA GEDEFA

Reviewer #3: No

Reviewer #4: **Yes: ** Rukhsana Gul

Associate Professor

Obesity Research Center

College of Medicine

King Saud University

Riyadh, KSA

---

## [Author Response · Author response to Decision Letter 2]

15 Sep 2025

Response to reviewer

Dear Editor and Reviewers,

We sincerely appreciate the time and effort you have dedicated to reviewing our manuscript. Your insightful comments and constructive suggestions have been invaluable in enhancing the quality of our case report.

We have carefully considered each comment and have incorporated the necessary revisions into the manuscript.

Below, we provide detailed responses to the reviewers’ comments.

Reviewer #2:

1. Line 408-410: Our study provides important insights into the CVD landscape in Hormozgan Province, revealing a prevalence of 7.8% (CI=0.071-0.084), with IHD accounting for 6.7% (0.044-0.09) and stroke for 2.2% (0.018-0.025)

1.1. Confidence intervals (CI) should be reported in the same unit as the main value. If you report prevalence as a percentage (e.g., 7.8%), then the CI should also be in percentage (CI: 7.1%–8.4%).

Authors response: Thank you for this helpful suggestion. We have revised the manuscript so that all confidence intervals are reported in the same units as the corresponding main values.

Page 24, line 391-392.

1.2. Discuss this finding (prevalence of CVD) by comparing and contrasting prevalence of CVD from other parts of the world. This helps the reader whether prevalence of CVD in the region is high or low compared to other parts of world.

Authors response: We sincerely thank the reviewer for their insightful and constructive comments. All the points requested have been thoroughly addressed and incorporated into the discussion section of the manuscript.

Page 24, line 392-395

2. Discussion line 445-451

Our logistic regression analysis …. increased age, increased BMI, hypercholesterolemia, diabetes mellitus, HTN, and severe anxiety…. include female sex

Hypertension and severe anxiety was discussed but increased age, increased BMI, hypercholesterolemia, diabetes mellitus and sex should be also discussed by comparing and contrasting with previous studies (whether your finding is similar or different)

Authors response: We are grateful for the reviewer’s attention to detail. Age, BMI, hypercholesterolemia, diabetes mellitus, and sex were discussed by comparing and contrasting our findings with those of previous studies.

Page 25-26, line 420-442

3. Line 463-465 and 471-472: Our study further emphasizes tailoring interventions, particularly to increase physical activity among women and urban populations….no finding in your study that suggest physical activity in women and urban population associated to CVD. Remove it

Authors response: Thank you for your constructive feedback. Following the reviewer’s suggestions, we have removed the items referenced in the comments and revised the manuscript accordingly.

Page 27, line 462-463 and line 455-456

4. Line 469-70: even though prevalence of obesity is high, no finding in your study that suggest obesity associated factor to CVD: better to remove it.

Authors response: Thank you for this helpful suggestion. Following the reviewer’s suggestions, we have removed the items referenced in the comments and revised the manuscript accordingly.

Page 27, line 461

5. Conclusion

Focus on your objectives

example like this. The prevalence of CVD was 7.8%(is it high/low comparing…). Factors significantly associated with CVD included older age, higher BMI, hypercholesterolemia, diabetes mellitus, hypertension, and severe anxiety. Interestingly, female sex appeared to be protective against CVD compared to male sex.

Authors response: We sincerely thank you for your constructive feedback, which has been invaluable in improving the quality of our article. We have implemented the referee’s suggestions and corrected the conclusion.

Page 28, line 486-489

---

## [Decision Letter · Decision Letter 2]

7 Oct 2025

Exploring Cardiovascular Disease Prevalence and Contributing Factors among Adults in Southern Iran, a Cross-Sectional Survey: Rationale, Design, and Primary Results

PONE-D-25-24170R2

Dear Dr. Abbaszadeh,

We’re pleased to inform you that your manuscript has been judged scientifically suitable for publication and will be formally accepted for publication once it meets all outstanding technical requirements.

Kind regards,

Amin Mansoori

Academic Editor

PLOS ONE

Additional Editor Comments (optional):

Reviewers' comments:

Reviewer's Responses to Questions

**Comments to the Author**

Reviewer #2: All comments have been addressed

2. Is the manuscript technically sound, and do the data support the conclusions?

Reviewer #2: Yes

3. Has the statistical analysis been performed appropriately and rigorously?

Reviewer #2: Yes

4. Have the authors made all data underlying the findings in their manuscript fully available?

Reviewer #2: Yes

5. Is the manuscript presented in an intelligible fashion and written in standard English?

Reviewer #2: Yes

Reviewer #2: In abstract line 29 6289 individuals and line 35 says 6,298 participants....the number of participants should be same==>6298

**Do you want your identity to be public for this peer review?** For information about this choice, including consent withdrawal, please see our Privacy Policy

Reviewer #2: **Yes: ** Debela Ejeta Gedefa

---

## [Editor Report · Acceptance letter]

PONE-D-25-24170R2

PLOS ONE

Dear Dr. Abbaszadeh,

I'm pleased to inform you that your manuscript has been deemed suitable for publication in PLOS ONE. Congratulations! Your manuscript is now being handed over to our production team.

Kind regards,

on behalf of

Dr. Amin Mansoori

Academic Editor

PLOS ONE